

# Domestic sewage dispersion scenarios as a subsidy to the design of urban sewage systems in the Lower Amazon River, Amapá, Brazil

Carlos Henrique Medeiros de Abreu[1], Elizandra Perez Araújo[2], Helenilza Ferreira Albuquerque Cunha[3], Marcelo Teixeira[4] and Alan Cavalcanti da Cunha[5]

[1] Environmental Engineering School, Amapá State University, Macapá, Amapá, Brazil
[2] Graduate Program in Biodiversity and Biotechnology, Federal University of Amapá, Macapá, Amapá, Brazil
[3] Environment and Development Department, Federal University of Amapá, Macapá, Amapá, Brazil
[4] Faculty of Civil and Environmental Engineering, Federal University of Pará, Tucuruí, Pará, Brazil
[5] Civil Engineering Department, Federal University of Amapá, Macapá, Amapá, Brazil

Corresponding author
Carlos Henrique Medeiros de Abreu,
carlos.abreu@ueap.edu.br

## ABSTRACT

The final in natura discharge of urban domestic sewage in rivers in the Amazon is a widespread practice. In addition, there is an evident lack of knowledge about the self-depurative characteristics of the receiving water bodies in these rivers. This problem is a challenge for designing sanitary sewage system (SSS) projects in the region. We aimed to numerically simulate hydrodynamic scenarios to study pollutant dispersion processes in an urban stretch impacted by domestic sewage in the Lower Amazon River (Amapá, Brazil) using a hydrodynamic model calibrated and coupled to a dispersive model (Lagrangian) (SisBaHiA). The following methodological steps were performed: (a) bathymetric and liquid discharge experimental campaigns using acoustic techniques (acoustic doppler current profiler—ADCP); (b) identification of point and diffuse sources of pollution in the Santana Channel (CSA) and North Channel of the Amazon River (NCM) in Macapá; (c) calibration of the hydrodynamic model and simulation of the dispersive process of domestic sewage plumes; (d) simulation of dispersive process scenarios in two seasonal hydrological periods and different tidal phases. The results of the simulations indicated significant spatiotemporal variations in the plumes, suggesting critical restriction of water quality in the dry period. The hotspot water collection supply station for ETA-CAESA was found to be the most threatened site by diffuse and point source loads. The simulated impacts showed that concentration variation worsens seasonally, restricting the multiple uses of water in both seasonal periods, regardless of tide phase. The pollutant plumes near the coastal-urban zone were apparently more inhibited by the influence of currents, and, due to the greater dilution capacity in the center of the channel, by the effect reversing with the approximation to the riverbank. The research hypotheses were supported: (a) the process of self-depuration of pollutants in the NCM has considerable limitations in shallow areas, and (b) SSS design projects in the region of the Amazon estuarine complex require hydrodynamic and strict water quality assessment, especially when their hydrological-seasonal and bathymetric characteristics are significantly unfavorable to dispersive

processes. Thus, a hydrodynamic analysis should be the primary criterion in designing any SSS projects in this stretch of the estuarine Amazon region.

## INTRODUCTION

Of the five Brazilian geographic regions (North, South, Southeast, Northeast, and Central-west), the North, including the estuarine Amazon region, has the worst sewage rate. On average, it has only 14% of the overall index of sewage access. For example, Macapá (Amapá state) is served by only 10.7%, and Porto Velho (Rondônia state) is served by 4.78%, both leading the worst national sewage service rates (*Sistema Nacional de Informações sobre Saneamento (SNIS), 2020*).

The disorderly urbanization process in the Amazonian estuarine region has negatively affected basic sanitation indicators, a persistent problem that severely affects public health (*Araújo et al., 2021*). This scenario also historically reflects the lack of investment in the sanitation sector throughout the national territory. For example, only five Brazilian capitals have water service supply rates greater than 90%, and only three capitals treat more than 80% of the collected sewage (*Sistema Nacional de Informações sobre Saneamento (SNIS), 2020*; *Viegas et al., 2021*).

Amapá has 16 municipalities; Macapá and Santana, the largest urban centers, are the focus of this research. In Macapá and Santana, the treated sewage service levels vary between 3 and 18.74%, and the remainder is dumped *in natura* in the rivers or water bodies that drain both municipalities (*Sistema Nacional de Informações sobre Saneamento (SNIS), 2020*; *Sousa, Cunha & Cunha, 2021*; *Viegas et al., 2021*), where sewage collection networks are mainly located and are generally old, inefficient, and obsolete.

The main consequence of the lack of planning and infrastructure for collection, distribution, and treatment of domestic sewage in the region is the constant threat of the impacts of *in natura* sewage on public health and on coastal ecosystems and water resources (*Cunha et al., 2001*; *Cunha et al., 2004*; *Cunha et al., 2005*; *Cunha et al., 2012*; *Sousa, Cunha & Cunha, 2021*).

As is often the case in the Amazon region, riverbanks in Macapá and Santana have not been the subject of much study or monitoring; therefore, little is known about the spatiotemporal variations of pollutant plumes in natural watercourses. In addition, studies addressing the variation in water quality and its environmental and sanitary compliance are remarkably scarce in the literature (*M. de Abreu et al., 2020*), and this is a persistent and growing scientific challenge. For example, the use of hydrodynamic field experimentation combined with numerical modeling and simulation is necessary to provide a rational, efficient, and crucial basis for planning and decision-making for proper management of

water bodies and their interactions with sanitation and the environment (*Chang et al., 2015*; *Altenau et al., 2017*; *Li, Mao & Li, 2017*).

In this context, the present study raised two research hypotheses: (1) the lack of knowledge of self-depurative capacity and dispersive processes (diffuse and point) in the bodies of water are aggravating factors that hinder management and decision-making of water resource management in the coastal zone; (2) the lack of planning, inadequacy and the necessary expansion of the urban sanitary sewage system (SSS) infrastructure preclude the necessary legal compliance with sanitary parameters in space and time (LAW 14,026/2020); (3) the spatiotemporal variations of the concentrations of pollutant plumes (dispersion) are seasonally influenced (rainy and less rainy seasons in the Amazon) by the geometry and bathymetry of the channel, impacting coastal zones beyond the limit distances used as legal references along the Macapá (RF-300 m) and Santana riverbanks (RF-600 m).

In this context, we aimed to develop hypothetical numerical scenarios for the dispersion of pollutant plumes (fecal coliforms; colony-forming units—CFUs) in urban stretches representative of the Macapá and Santana riverbanks. The sanitary sewage load was adopted in a dispersive model (Lagrangian) coupled with a previously calibrated hydrodynamic model, allowing it to represent the diffusive-advective behavior of sanitary sewage plumes in the coastal area of interest. The scenarios resulting from the simulations represent basic contributions to SSS or alternative design studies, such as underwater emissaries that are eventually connected to primary sanitary sewage treatment plants (*Velz, 1984*; *Von Sperling, 2017*).

## MATERIALS AND METHODS

### Study area

The dynamics of the pollutant dispersion was assessed in a representative section of the North Channel of the Amazon River (NCM), between the Vila Nova River and the Igarapé (= creek) do Curiaú. The Santana Channel is located along this stretch of Santana, partially confined by Santana Island and the edge of Macapá, to the Igarapé do Curiaú (Fig. 1). The channel comprises a distance of 50 km in the coastal estuarine region of Amapá (*Torres, El-Robrini & Costa, 2018*).

In Santana, Companhia das Docas de Santana (CDSA) is equipped with a port area that exports commodities (manganese, wood, eucalyptus, and cellulose, among others) to and from other Amazonian regions. The navigation route is unique for vessels of up to 50 thousand tons through the NCM (*Pereira et al., 2014*; *Cunha et al., 2021*; *Araújo et al., 2022*) and is considered a sanitary risk area close to the urban areas of Macapá and Santana. For example, ships' ballast water in this zone is considered an environmental and sanitary threat to biological resources because it is an import zone of ballast water (*Pereira et al., 2014*). This zone also includes an increased risk of oil and derivatives spills (*Cunha et al., 2021*; *Araújo et al., 2022*); therefore, it is an area of significant economic and environmental interest in water resource management, biodiversity conservation, and environmental sanitation.

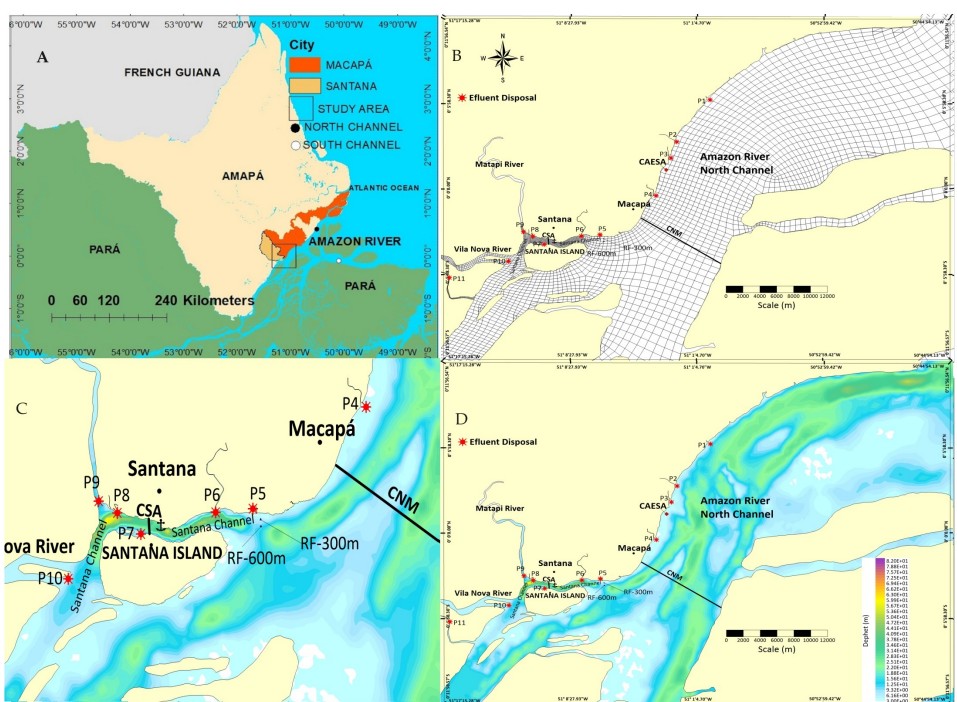

**Figure 1** (A) Geographic location; (B) Numerical mesh of finite elements; (C) Details of the Santana Channel (CNS–internal coast protected by Santana Island; (D) North Channel of the Amazon River (Macapá–NCM) (external coast) with bathymetry (m) and effluent discharge points. Legend: (P1) Igarapé do Curiaú; (P2) Channel of Jandiá; (P3) Igarapé das Mulheres; (P4) Igarapé das Pedrinhas; (P5) Fazendinha Environmental Protection Area; (P6) Igarapé da Fortaleza; (P7) Island of Santana; (P8) Bairro do Elesbão—STN; (P9) Matapi River; (P10) Vila Nova River; (P11) Beija-Flor River. Figure source credit: *M. de Abreu et al. (2020)*, CC BY 4.0.

Hydrodynamically, the study stretch is considered fluvial (≈250 km upstream of the North Atlantic Ocean), maintaining freshwater quality, but significantly influenced by variations in semidiurnal estuarine tides and with no detectable saline intrusion (*Ward et al., 2013*). CDSA tide gauges indicate that the amplitude or water level can vary up to ≈3 m (meso tides), favoring two daily reversals of the Amazon River currents (*Less et al., 2021*).

This coastal estuarine region can be subdivided into high, medium, and low stretches and is situated in Macapá Bay. The plain of this bay is often interrupted by tertiary formations (Barreiras Group sediments) similar to cliffs, with meanders, residual lakes, and "understories". The term "understory is related to coastal wetlands, which consist of lagoons and lakes exclusively found in urban areas of Macapá and Santana, influenced or not by the tide, in addition to mangroves that make up the ecosystems of the Macapá bay (*Torres, El-Robrini & Costa, 2018*).

## Climatic data

The climate of the region is classified as Am (monsoon equatorial), with elevated temperatures (never lower than 18 °C) and a pronounced drought period (August to November) (*Köppen, 1936*). The climatic variation occurs in the Amazon rainy season

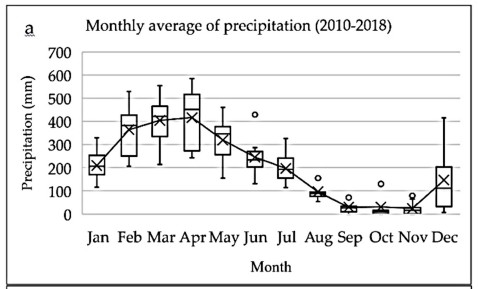
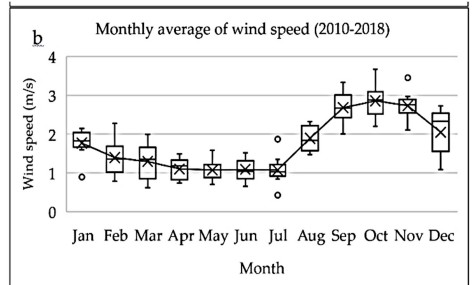

**Figure 2** **(A) Average monthly rainfall (mm) and (B) intensity of the average monthly wind speed (m/s) (2010–2018)—source: BDMEP–INMET.** Figure source credit: *M. de Abreu et al. (2020)*, CC BY 4.0.

(December to August), with maximum rainfall between March and May, and less rainy Amazon season (September to November), with minimum rainfall in October (*Limberger & Silva, 2016*; *Marengo & Espinoza, 2016*; *De Souza et al., 2017*).

The pattern of seasonal variation in wind intensity follows a behavior opposite to that of rainfall (Figs. 2A and 2B), meaning it is less intense in the rainy season and more intense in the less rainy season (2010–2018 series). This series was obtained from the meteorological station of Macapá (MACAPA-AP-OMM: 82098)—Meteorological Database for Teaching and Research (Banco de Dados Meteorológicos para Ensino e Pesquisa—BDMEP) of the National Institute of Meteorology of Brazil (Instituto Nacional de Meteorologia do Brasil—INMET).

## Experimental determination of liquid discharge (CSA and NCM) and bathymetry between the Igarapé do Curiaú and the Vila Nova River

The stretch that includes the CSA and NCM between the Igarapé do Curiaú (P1) and the Beija-Flor River (P11) was selected to evaluate the dynamics of natural coastal flow under the influence of large bodies of water and similar boundary and initial conditions in open channels.

Experimental campaigns of liquid discharge quantification were performed in the CSA and NCM sections to calibrate the hydrodynamic model. The procedure was performed with an acoustic method using an acoustic doppler current profiler (ADCP, Surveyor M9; SonTek). The ADCP was installed on the side of a boat measuring 22 m long, with the transducers immersed 1.0 m below the water surface. Compass calibration with GPS was performed at the beginning of each field session (*M. de Abreu et al., 2020*).

In the CSA section, the campaign was carried out between the neap and low spring tides on March 18, 2019, from 6:00 a.m. to 6:30 p.m., comprising a complete semidiurnal tide cycle, with 133 crossings in the channel section, whose average width is ≈650 m. In the NCM section, the campaign was carried out on November 7, 2017 (less rainy season) and March 19, 2019 (rainy season), in 12 crossings lasting approximately one hour each (≈12.5 km) (Fig. 1D).

The refined bathymetric survey of the margin, later incorporated into the hydrodynamic model, was carried out between P$_1$ and P$_{11}$ on November 8, 2017 (Fig. 1B). The procedure

started at 9:00 a.m. in front of Igarapé do Curiaú and ended at 5:00 p.m. in front of the Vila Nova River, on a "zigzag" trajectory along the bank (Figs. 1C and 1D).

## Modeling and hydrodynamic simulation and dispersion of sanitary effluents

SisBaHiA (*Rosman, 2018*) was used to generate the analysis of predicted tide and hydrodynamic simulation (May, August, and November) and effluent dispersion (May and November) between the Igarapé do Curiaú and Vila Nova River. This software has a 3D or 2DH hydrodynamic circulation model optimized for natural water bodies, to which Eurelian and/or Lagrangian models can also be integrated to describe pollutant transport phenomena (*Rosman, 2018*). In this study, we opted for the 2DH model (Eqs. (1)–(3)), which represents the hydrodynamic behavior. The Lagrangian model was considered the most suitable for this purpose, represented by the following equations:

$$\frac{\partial U}{\partial t} + U\frac{\partial U}{Ux} + V\frac{\partial U}{\partial y} = -g\frac{\partial \zeta}{\partial x} - \frac{gH}{2\rho_\circ}\frac{\partial \hat{\rho}}{\partial x} + \frac{1}{H\rho_\circ}\left(\frac{\partial (H\hat{\tau}_{xx})}{\partial x} + \frac{\partial (H\hat{\tau}_{xy})}{\partial y}\right)$$
$$+ \frac{1}{H\rho_\circ}\left(\tau_x^S - \tau_x^B\right) - \frac{1}{H\rho_\circ}\left(\frac{\partial S_{xx}}{\partial x} + \frac{\partial S_{xy}}{\partial y}\right) + 2\Phi sen V - \frac{U}{H}\Sigma q \tag{1}$$

$$\frac{\partial V}{\partial t} + U\frac{\partial V}{Ux} + V\frac{\partial V}{\partial y} = -g\frac{\partial \zeta}{\partial y} - \frac{gH}{2\rho_\circ}\frac{\partial \hat{\rho}}{\partial x} + \frac{1}{H\rho_\circ}\left(\frac{\partial (H\hat{\tau}_{xy})}{\partial x} + \frac{\partial (H\hat{\tau}_{xy})}{\partial y}\right)$$
$$+ \frac{1}{H\rho_\circ}\left(\tau_y^S - \tau_y^B\right) - \frac{1}{H\rho_\circ}\left(\frac{\partial S_{yy}}{\partial y} + \frac{\partial S_{xy}}{\partial x}\right) - 2\Phi sen U - \frac{V}{H}\Sigma q \tag{2}$$

$$\frac{\partial \zeta}{\partial t} + \frac{\partial UH}{Ux} + \frac{\partial VH}{\partial y} = \Sigma q. \tag{3}$$

In the two equations, U is defined as the velocity on the $x$-axis (m/s); V is the velocity on the $y$-axis (m s$^{-1}$);—$\zeta$ represents the free surface elevation (m); H is the depth of the water column (m); $\rho_o$ is the average density of water (kg m$^{-3}$); $\rho_r$ is the density of the reference water (kg m$^{-3}$); g is the acceleration of gravity (m s$^{-2}$); $\tau_{ij}$ is the turbulent stress tensor; (i,j) represents the indices on the horizontal (x; y) plane; $\tau$_iS and $\tau$_iB are the wind stresses on the water surface, and the lower friction stress, respectively (kg m$^{-1}$s$^{-2}$); 2 $\Phi$sen $\theta$U and 2 $\Phi$ $\theta$senV represent the Coriolis accelerations; $\Phi$ is the angular rotational velocity of the Earth (rad s-1); $\theta$ is the latitude angle; $\sum$q represents the water balance that considers atmospheric rainfall, infiltration, and evaporation; S$_{ij}$ represents the effect of the radiation stresses (*Cunha et al., 2006*; *Rosman, 2018*).

In the Lagrangian model of SisBaHiA, contaminant sources are represented by number of particles released in the source region at regular time intervals. The randomly arranged particles in the source region are then carried by the currents computed by the hydrodynamic (calibrated) model. The position of any particle at the next instant is defined

by $P^{n+1}$, approximated by a second-order Taylor Series expansion. $P^{n+1}$ iscalculated from the previously known previous position ($P^n$) (Eq. (4)):

$$P^{n+1} = P^n + \Delta t \frac{dP^n}{dt} + \frac{\Delta t^2}{2!} \frac{d^2 P^n}{dt^2} \qquad (4)$$

For effluents discharged from any source, the amount of mass ($Ma$) of a given species (*e.g.*, biological oxygen demand—BOD or CFU) is represented by a particle entering the modeled domain, which is given by Eq. (5) (*Rosman, 2018*):

$$M_a = \frac{QC_a x \Delta \tau}{N_P}. \qquad (5)$$

$Q$ represents the effluent discharge from the source, $Ca$ is the concentration of the substance $a$ present in the discharge from the source, and $NP$ is the number of particles entering the domain per time step $\Delta t$. The dimensions of the source region are defined in such a way that the concentration in the source region is equal to that observed at the end of the initial dilution process, that is, within the near field of the mixture of the contaminant plume (*Rosman, 2018*).

## First-order kinetics of the Lagrangian model

SisBaHiA considers that the total amount of effluent load, $Q_T$, discharged in a given source region per time interval $\Delta \tau$, is defined by:

$$QT = QQ/s \times \Delta \tau \qquad (6)$$

If $NP$ particles are launched in each $\Delta \tau$, the initial amount of each particle, $m_o$, will be:

$$m_o = \frac{QT}{N_P} = \frac{QQ/s \times \Delta \tau}{N_P} \qquad (7)$$

Thus, it is assumed that over time, the remaining amount in each particle, $m_{(tv)}$, is a function of its lifetime, "$tv$". That is, it is possible to specify kinetic reactions, $R_{(tv)}$, that alter the initial amount of each particle as follows:

$$m(t_v) = m_o R(t_v) \qquad (8)$$

First-order kinetic reactions (exponential decay) may be conveniently specified with the prescription of parameter $T_{90}$. That is, the time required for the decay of 90% of the value of the initial concentration ($m_o$) or reduction of an order of magnitude. In this way, the kinetic reaction is given as:

$$R(t_v) = exp(-k_d t_v) \qquad (9)$$

Where the reaction constant $K_{d(decay)}$ is calculated as a function of $T_{90}$ as:

$$K_d = -\ln \left( \frac{0.1}{T_{90}} \right) \qquad (10)$$

## Initial contour conditions and the Lagrangian model

We can use two types of boundary conditions in SisBaHiA: 1) closed boundary nodes for the imposition of water discharge values and 2) open boundary nodes for free surface elevation values (*Cunha et al., 2006*) (Figs. 1B and 1D).

In the closed limit, a net discharge of the Amazon River was imposed (collected at the Óbidos station and experimental data in the NCM), and two of its main tributaries (the Xingu and Tapajós rivers) based on a measurement campaign carried out at the CSA and NCM, previously described or on experimental data available in the literature (*Silva, 2009*; *Barros & Rosman, 2018*), such as those of the National Water Agency (*Agência Nacional de Águas (ANA), 2017*) of the Óbidos station (code 00155001). In addition, rainfall levels, direction, and wind intensity were obtained from the Macapá weather station.

In the open limit, a series of water surface elevations was imposed to represent the tidal waves considering the six main tidal components (M2, S2, N2, K1, O1, and M4) imposed on nodes of the open limit mesh (Fig. 1B).

In the Lagrangian model, effluent discharge points (Fig. 1C), estimated raw sewage flow, and their respective concentrations were defined (Table 1). In addition, due to its environmental and sanitary importance to the coast, control points or imaginary reference lines were defined along the coast (RF at 300 m away from the riverbanks—RF-300; RF at 600 m away from the river banks RF-600), in addition to the point at the water collection station (CAESA) of Macapá. For example, RF-300 has been commonly used as a precautionary and safety line in outfall projects (RF-300 m) (*Metcalf & Eddy, 1991*; *Rodrigues, 2012*). Justification for this choice is the prior knowledge that displacement of pollutant plumes is concentrated in this zone and that there is a tendency to remain close to the riverbanks and for long distances greater than 50 km (*Cunha et al., 2021*; *Araújo et al., 2022*).

The choice of effluent launch points was based on an experimental campaign (November 2017), and Sewage Atlas reports available on the website of the National Water Resources Information System (Sistema Nacional de Informações sobre Recursos Hídricos—SNIRH) and the National Water Agency (*Agência Nacional de Águas (ANA), 2017*). These reports define specific raw sewage launch points (Figs. 1C and 1D) for what is known as the Metropolitan Zone of Macapá—MZM: Macapá (Igarapé do Curiaú), Santana (Vila Nova River), and Mazagão (Beija-Flor tidal channel) according to their population numbers.

A maximum concentration value of $10^8$ (CFU/100 mL of the sample) was used for the pollutant (thermotolerant CF or CFU) defined at the launch points imposed on the model. For CFU, the lifetime (T) or decay time of 90% of the particles ($T_{90}$) equal to $T_1$ of 21,600 s (6 h) was defined (*Barros et al., 2015*; *Von Sperling, 2017*). In addition, simulations were performed for $T_{90}$ equal to $T_2 = 16,200s$ (4.5 h) and $T_3 = 27,000s$ (7.5 h) to analyze sensitivity, self-depuration capacity, dynamics of pollutant dispersion in the analysis sections in NCM and CSA, and variations of the results in relation to decay.

## Computational mesh

The computational domain covers an area of approximately $3 \times 10^7$ km$^2$, containing 2,238 homogeneously distributed elements and totaling 10,627 nodes (Figs. 1B and 1D). The

**Table 1** Effluent release points, concentration, and flow rate imposed on the model estimated by the National Water Agency (Agência Nacional de Águas - ANA, 2020).

| Point | Location reference | CFU Concentration Most probable number/100 mL per sample | Estimated flow Q (m³ s⁻¹) |
|---|---|---|---|
| P1 | Igarapé do Curiaú | $10^8$ | 0.197 |
| P2 | Channel of Jandiá | $10^8$ | 0.158 |
| P3 | Igarapé das Mulheres | $10^8$ | 0.158 |
| P4 | Channel of Pedrinhas | $10^8$ | 0.158 |
| P5 | Channel of Paxicu | $10^8$ | 0.002 |
| P6 | Igarapé da Fortaleza | $10^8$ | 0.400 |
| P7 | Santana launch point | $10^8$ | 0.002 |
| P8 | Santana launch point | $10^8$ | 0.002 |
| P9 | Matapi River | $10^8$ | 0.010 |
| P10 | Vila Nova River | $10^8$ | 0.004 |
| P11 | Beija-Flor channel (Mazagão) | $10^8$ | 0.010 |
| | Sum of Loads | $10^8$ | 1.101 |

bathymetric data inserted in the mesh were obtained from the nautical charts of the Brazilian Navy [Directorate of Hydrography of the Navy (Diretoria de Hidrografia da Marinha—DHN); Hydrography Center of the Navy (Centro de Hidrografia da Marinha"—CHM)] (*Ministério da Defesa do Brasil/Marinha do Brasil, 2019*). They were then integrated with local experimental data and refined along the coast near Macapá and Santana, between the Igarapé do Curiaú and Vila Nova River (November 2017), generating approximately 19,000 additional points in the mesh.

## Model calibration and verification in estimating pollutant dispersion

Data of the tidal rise (tide gauge station—CDSA) and the hydrodynamic data measured with those simulated by the model were compared to calibrate the hydrodynamic model. Flow and velocity measurements performed in the field (ebb and flood) were statistically compared with the model results, observing the acceptable error ranges of the parameters. When the errors exceeded acceptable statistical limits, adjustments to the parameters were made until their calibration (*Hsu et al., 1999*; *Liu et al., 2008*; *Fossati & Piedra-Cuevai, 2013*; *M. de Abreu et al., 2020*; *Cunha et al., 2021*; *Araújo et al., 2022*).

At the boundary of the open limit in the computational domain region, the harmonic constituents (M2, S2, N2, K1, O1, and M4) were considered and observed in the study region after the analysis of tidal elevations (Figs. 1B and 1D). The value of the constituents was imposed on the nodes of the open limit using adjustment of coefficients (in amplitude and phase) for each harmonic constituent (*Liu et al., 2008*; *Nzualo, Gallo & Vinzon, 2018*).

For model calibration and due to quality and consistency, the tidal rise data of 2016 from the CDSA were used as a reference. The objective was to reproduce the same harmonic components of the local tide with the smallest possible error and use the tide projection of SisBaHia. Thus, the tidal behavior predictions for 2019 were made. To estimate the

reliability of the projections, comparisons were made between data projected by the model and those observed in 2017 and 2018.

## Statistical analysis

The significance level evaluated between the experimental results and those of the predicted tides generated in SisBaHiA was tested, comparing the tidal levels recorded in March and November 2017 and May and August 2018. The modeled hydrodynamic results for May, August, and November 2019 were compared with the same corresponding predicted tides for 2019. Thus, flows between the tidal cycles (experimental and simulated) were statistically tested by successive comparisons with results simulated by the model for March 2019.

Quantification of the efficiency of the mathematical models was performed using the Nash-Sutcliffe (NSE) methods (Eq. (11)), Pearson's correlation, $R^2$, and concordance index (d) (Eq. (12)). These statistical indicators were selected according to their different practical applications in hydrodynamic modeling, such as testing the accuracy and ability of models to represent physical reality (*Devkota & Fang, 2015*; *Bakken, King & Alfredsen, 2016*; *Skhakhfa & Ouerdachi, 2016*; *Haddout & Maslouhi, 2018*).

$$E_{NS} = 1 - \frac{\sum_{i=1}^{n}(O_i - P_i)^2}{\sum_{i=1}^{n}(O_i - \overline{O})^2} \tag{11}$$

$$d = 1 - \frac{\sum |P - O|^2}{\sum (|P - \overline{O}| + |O - \overline{O}|)^2} \tag{12}$$

where $O_i$ and O represent the observed tidal rises and $P_i$ and P represent the values predicted by the model. The results of the model efficiency can vary between 1 and $-\infty$ in Eq. (11) (Nash-Sutcliffe efficiency), where 1 indicates the model that perfectly represents the observed data. Values greater than 0.75 indicate that the model is capable of representing what was experimentally observed (*Al-Asadi & Duan, 2015*).

Similar to the NSE efficiency test, the concordance index test (d) compares data generated by the model (P) with those observed (O): if $d = 1$, the model is considered perfectly coherent; if $d = 0$, the model shows a lack of ability to represent the reality of the observed data set. The minimum acceptable value for d would be 0.75.

## RESULTS

### Experimental results in the Santana Channel(CSA) and North Channel of the Amazon River—Macapá (NCM)

The results of experimental flow measurements performed in CSA and NCM (March 2019) and NCM (November 2017) were represented by the liquid discharge curves (Supplement-S1). In March 2019, the maximum ebb value in the CSA section was 22,729 $m^3s^{-1}$, and the maximum flood value was $-13,381$ $m^3s^{-1}$. In the rainier period (March 2019) in the NCM, values higher than those measured in November were obtained, with a maximum of 254,944 $m^3s^{-1}$ in the ebb and $-149,653$ $m^3s^{-1}$ in flood. In the less rainy season (November 2017),

the values of 208,637 m$^3$s$^{-1}$ (ebb) and $-$195,307 m$^3$s$^{-1}$ (flood) were obtained, respectively. For both channels, the reversal period (in flood) was significantly shorter (between 3 h 20 min and 3 h 40 min) than the ebb period (between 8 h 20 min and 8 h 40 min).

The detailed velocity profiles of the currents measured with the ADCP, CSA (Supplement-S2a), and NCM (Supplement-S2b) are naturally interconnected (Fig. 1D). Therefore, they present interdependent hydrodynamic and bathymetric characteristics. The different colors of each vertical profile indicate vertical and transverse variations of the instantaneous velocities along the sections of each channel. When these profiles are integrated, it is possible to quantify maximum speed values in channels up to $\approx$2 m s$^{-1}$ (light and warmer colors).

The characteristics of the channels are very different, with average widths of $\approx$ 600–700 and depths of 20–35 m to CSA. The average width of the NCM is 12.3 km, and depth varies between 10–38 m. In the NCM section, three "sub-channels" normally influence the currents' lateral, transverse, and longitudinal distribution, resulting in a high complexity of the 3D hydrodynamic flow profile.

## Predicted observed tide and measured and simulated liquid discharge

The SisBaHiA predicted model provided the estimated tidal rise data (May to November 2019). To confirm the accuracy of the data predicted by the model, we performed statistical efficiency tests to compare the data observed at the CDSA station in 2017 and 2018 (*M. de Abreu et al., 2020*) (Table 2).

The statistical tests indicated reliable comparative results at the CDSA station and were used to reproduce numerical tidal elevation outputs without needing experimental data for March, May, August, and November 2019. Subsequently, these results were statistically compared with the increase in the tide in CDSA, which was used as a local tide gauge reference, and the levels generated by the hydrodynamic model of SisBaHiA.

The analysis indicated that the hydrodynamic model satisfactorily corresponds to the tidal elevations of the CDSA tide gauge station. Therefore, we correlated the efficiency of the simulated hydrodynamic model (variation of the experimental liquid discharge) of March 2019, both in the CSA and the NCM calibration section. The adjustment values of R$^2$ for the simulated and observed hydrodynamic behavior were greater than 0.95. The other statistical tests (Pearson, "*d*", and NSE) for 03/18/2019 and 03/19/2019 indicated values higher than 0.9. The same occurred for values of experimental and simulated flood flow and velocity intervals with regard to CSA (03/18/2019) and NCM (03/19/2019) (Table 3).

## Hydrodynamic simulation model

In simulations of the hydrodynamic model for the CSA and NCM sections in 2019, the moments of maximum flow (May) and maximum flood (November) were selected as references for simulating pollutant dispersion using the Lagrangian Model during the spring tide (Supplement-S3a–S3b).

The model-derived current values in NCM varied between 0.134 m/s and 3.34 m/s, with peak values concentrated in the center of the channel. In the CSA, speeds ranged from 0.133 m/s to 1.47 m/s, with higher speeds observed throughout the reference section (Supplement-S4).

**Table 2** Statistical analysis between tidal data observed, predicted, and simulated by SisBaHiA. Source: The authors, 2022.

| Period | Observed × Predicted ($p < 0.01^*$) | | | |
| --- | --- | --- | --- | --- |
| | Pearson correlation coefficient | Nash-Sutcliffe estimate (NSE) | $R^2$ | $d$ |
| March 2017 | 0.99 | 0.97 | 0.97 | 0.99 |
| November 2017 | 0.99 | 0.97 | 0.97 | 0.99 |
| May 2018 | 0.97 | 0.94 | 0.95 | 0.99 |
| August 2018 | 0.99 | 0.97 | 0.98 | 0.99 |
| Average-2017–2018 | 0.99 | 0.96 | 0.97 | 0.99 |

| Period | Predicted × Simulated ($p < 0.01^*$) | | | |
| --- | --- | --- | --- | --- |
| | Pearson correlation coefficient | Nash-Sutcliffe estimate (NSE) | $R^2$ | $d$ |
| March-2019 | 0.95 | 0.90 | 0.90 | 0.97 |
| May-2019 | 0.96 | 0.90 | 0.91 | 0.98 |
| August-2019 | 0.95 | 0.90 | 0.90 | 0.97 |
| November-2019 | 0.96 | 0.91 | 0.92 | 0.98 |
| Average-20189 | 0.96 | 0.90 | 0.91 | 0.98 |

**Table 3** Comparative results between simulated and experimental hydrodynamic parameters in the Santana Channel (CSA; 03/18/2019) and North Channel of the Amazon River (NCM; 03/19/2019). Source: The authors, 2022.

| Method/Parameter | Ebb flow rate ($m^3 \, s^{-1}$) ($\times 10^3$) | Flood flow rate ($m^3 \, s^{-1}$) ($\times 10^3$) | Average speed of the ebb current ($m \, s^-$) | Average speed of the flood current ($m \, s^{-1}$) |
| --- | --- | --- | --- | --- |
| Experimental-ADCP (CSA) | 22.73 | −13.38 | 0.98 | 0.55 |
| Simulated (CSA) | 22.41 | −12.36 | 1.13 | 0.59 |
| Relative error (%) (CSA) | 0.01 | 0.076 | 0.15 | 0.07 |
| Experimental-ADCP (NCM) | 254.94 | −149.65 | 0.98 | 0.72 |
| Simulated (NCM) | 249.94 | −143.92 | 1.40 | 0.75 |
| Relative error (%) (NCM) | 0.020 | 0.04 | 0.17 | 0.042 |

Similar analyses were conducted for November 2019 during both spring and flood tide. The current velocity values in the reference section during the flood in November varied between 0.138 m/s and 3.44 m/s in NCM and 0.25 m/s and 1.36 m/s in CSA, respectively (Supplement-S5).

## Dispersion and concentration of pollutants on the coast of Macapá and Santana

Spatiotemporal variations in the behavior of pollutant plumes showed a tendency of concentration and distribution along the Macapá and Santana shores. According to the Lagrangian model, after 1, 10, 20, and 30 days from the start of simulations, and considering a hypothetical decay rate of 21,600 s (May/maximum and November/minimum flows), the results of the dispersive processes of pollutants (total thermotolerant coliforms = CFU)

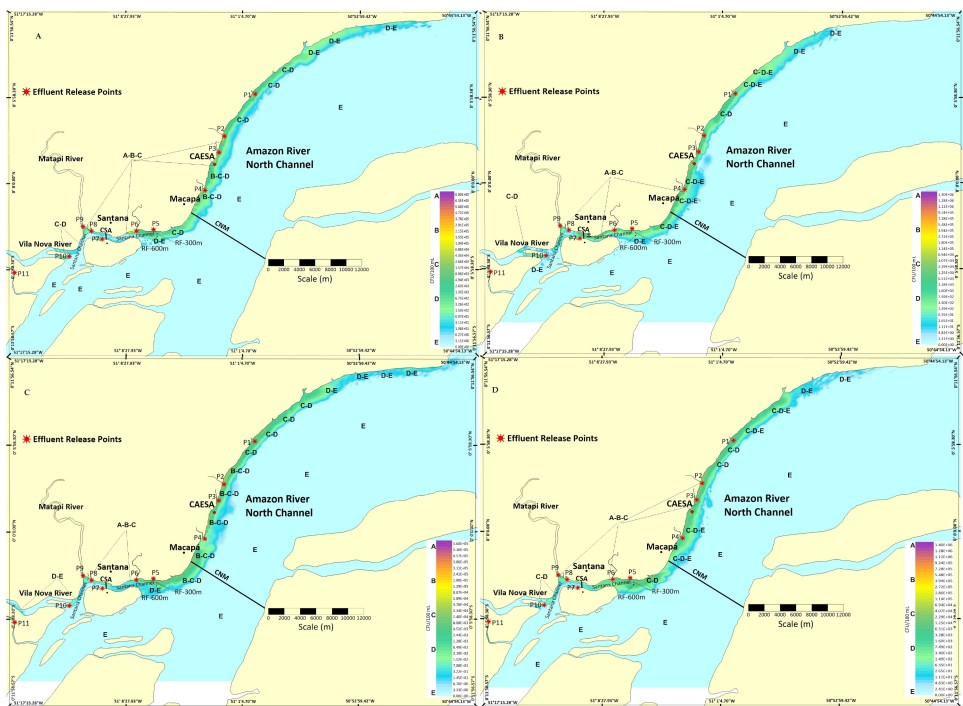

**Figure 3** **Dynamics and extent of the impact of the pollutant plume dispersion on the coast of Macapá and Santana.** (A) One day after the May/2019 release; (B) one day after the November release; (C) 10 days after the May release; (D) 10 days after the November release. Legend: (P1) Igarapé do Curiaú; (P2) Channel of Jandiá; (P3) Igarapé das Mulheres; (P4) Igarapé das Pedrinhas; (P5) Fazendinha Environmental Protection Area; (P6) Igarapé da Fortaleza; (P7) Island of Santana; (P8) Bairro do Elesbão–STN; (P9) Matapi River; (P10) Vila Nova River; (P11) Beija-Flor River.

are detailed in scalar fields (concentrations) (Figs. 3 and 4). Figures 3A–3B shows the fields of variation of the CFU concentration and geometric variation of the plumes for May (Fig. 3A) and November (Fig. 3B). The pollutant plume projected more toward the center of the channel in up to ≈2 km (NCM) in November compared to May.

On May 1st (Fig. 3C), the longitudinal extension of the plume was ≈63 km longer than in November. Despite being shorter longitudinally (≈52.2 km) (Fig. 3D), it tended to expand with more intensity laterally.

We found a greater expansion of the pollutant plume toward the center of the NCM ≈3 km wide between P1 (Igarapé do Curiaú) and P4 (downstream of the Igarapé das Pedrinhas) when compared to the 1st and 10th day after pollutant dispersion began (Figs. 4A and 4B). Spatially, the pollutant plume had higher concentrations near P5 (downstream of the Igarapé da Fortaleza) and between P2 (Channel of Jandiá) and P4 (upstream of the Igarapé das Pedrinhas), in addition to increasing its width by up to approximately 3.5 km in the transverse direction of the NCM.

Thirty days after pollutant dispersion simulations started, the increase in the CFU concentration level was also observed in May (Fig. 4C), and the maximum peak concentration value reached between all simulations was $1.9 \times 10^6$ CFU/100 mL. In

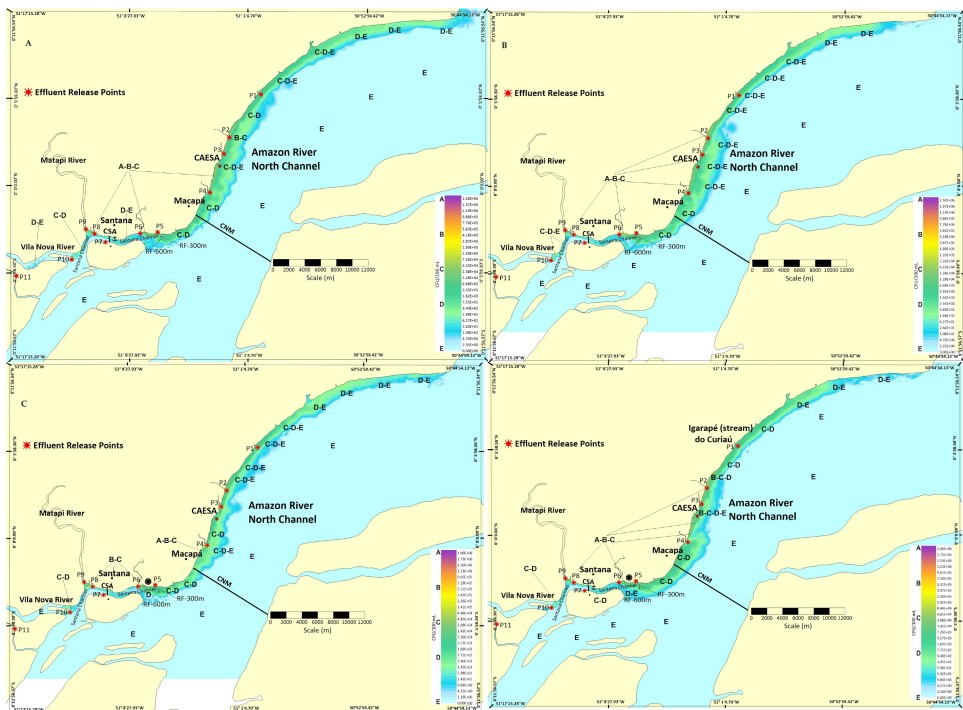

**Figure 4 Dynamics and extent of the impact of the pollutant plume dispersion on the coast of Macapá and Santana.** (A) Twenty days after the May launch; (B) 20 days after the November launch; (C) 30 days after the May launch; (D) 30 days after the November launch. Legend: (P1) Igarapé do Curiaú; (P2) Channel of Jandiá; (P3) Igarapé das Mulheres; (P4) Igarapé das Pedrinhas; (P5) Fazendinha Environmental Protection Area; (P6) Igarapé da Fortaleza; (P7) Island of Santana; (P8) Bairro do Elesbão—STN; (P9) Matapi River; (P10) Vila Nova River; (P11) Beija-Flor River.

the selected period, NCM and CSA were in the flood tide phase, and the pollutant plume had a greater extension area at P4 (Igarapé das Pedrinhas). Thirty days after the release of pollutants in November (Fig. 4D), the maximum concentration peak reached was $1.5 \times 10^6$. In November, this same order of magnitude was observed in all simulations.

Regarding P10 (Vila Nova River), the concentration values varied with the tide period, between 200 and 1,000 CFU/100 mL. However, at P11, concentrations remained above 2,000 CFU/100 mL between the Vila Nova River and the launch point without directly contributing to the pollutant plume released into the Amazon River.

Table 4 describes the variations in effluent concentration for the chosen points or along the coastal reference line (P-300, P-600, and CAESA), considering 1, 10, 20, and 30 days after the release of the pollutants. The criteria $T_{90} = 21,600s$, $27,000s$, and $16,200s$ were adopted, respectively.

At $T_1$, there were significant seasonal differences between the average concentration permanence values at the reference geographical imaginary points or lines for the two periods (May and November). The average decay time of CFU concentration in May presented 40% of the average value of November at the CAESA point (hotspot), 54% in relation to the RF-300, and 53% in relation to the RF-600. Spatially for $T_1$, the concentration

**Table 4 Descriptive synthesis of concentration variation of CFU/100 mL (* May; ** November).** Source: The authors, 2022.

| Point of reference | CFU/ 100 mL | | | | |
|---|---|---|---|---|---|
| | 1 day | 10 days | 20 days | 30 days | Medium |
| | | | $T_1 = 21600$ s (6.0 h) | | |
| Caesa | 2350*/8800** | 980*/4800** | 5000*/4195** | 1300*/6200** | 2407*/5998** |
| RF-300 (m) | 1380*/621** | 3860*/12129** | 9700*/9661** | 6848*/17656** | 5447*/10016** |
| RF-600 (m) | 480*/344** | 828*/5650** | 3297*/3504** | 1774*/2120** | 1549*/2904** |
| | | | $T_2 = 27000$ s (7.5 h) | | |
| Caesa | 2387*/21348** | 3163*/10294** | 9327*/11585** | 1704*/7022** | 4145*/13159** |
| RF-300 (m) | 3742*/1112** | 11191*/10163** | 6012*/54** | 15666*/14684** | 9152*/6503** |
| RF-600 (m) | 956*/387** | 4077*/4405** | 565*/354** | 5142*/3947** | 2685*/2273** |
| | | | $T_3 = 16200$ s (4.5 h) | | |
| Caesa | 2304*/7000** | 2950*/4363** | 1042*/7041** | 1163*/5043** | 1864*/5861** |
| RF-300 (m) | 815*/131** | 4424*/6004** | 10427*/2258** | 2160*/23852** | 4456*/8061** |
| RF-600 (m) | 430*/88** | 1635*/1214** | 2800*/2516** | 951*/920** | 1454*/1184** |

variations between the RF-300 and RF-600 references decreased significantly concerning the distance from the margins. The average of the RF-600 values represented only 28% of the average of the RF-300 values (Table 4).

For $T_2$, the differences between average concentrations in RF-300 and RF-600 behaved differently than in the $T_1$ simulations. For example, the CFU concentration averages obtained for RF-300 and RF-600 were higher in May (40%) than in November (18%) (Table 4).

At $T_2$ (27,00s), the concentration of CFU/100 mL in May had the highest averages compared to the $T_3$ and $T_1$ decay. In November, only the average concentration at the CAESA point was higher in relation to $T_3$ and $T_1$. Only the average concentration in the RF-300 reference in May, along with the $T_2$ decay time, was higher than in November when compared with $T_1$ and $T_3$ (Table 4).

Possibly due to a shorter decay time (in theory, with higher clearance rates), the average concentrations of $T_3$ (16,200 s) showed lower values than those observed for $T_1$ and $T_2$. As expected, they had similar seasonal and spatial behavior. In May, the CAESA reference presented only 40% of the concentration in November, and the RF-300 reference had 55% in relation to November. The behavior was different regarding the RF-600 reference, which displayed a concentration in May that was 22% higher than in November (Table 4).

High values of overall average concentrations in the respective references were found for $T_2$, with 6,319 CFU/100 mL. Still, similarly, the overall average concentration value was 4,720 CFU/100 mL for $T_1$ and 3,813 CFU/100 mL for $T_3$. As expected, the RF-600 reference had the lowest overall average pollutant concentration (Table 4). The RF-300 reference had the highest concentration for $T_1$ and $T_3$. Therefore, only for $T_2$ the overall average pollutant concentration was higher, at the CAESA point (Table 4).

## DISCUSSION

The variation in net discharge and current velocity fields is notably associated with tidal variation and hydrological seasonality. Thus, the similar behavior of both channels (CSA and NCM) is naturally explained, due to their natural hydraulic connection (Table 3).

We found a direct correlation between tidal variation and net discharge in sections CSA and NCM represented by the model ($R_{aj}$ >0.90). This suggests physical consistency and reliability of the hydrodynamic model, properly representing the mass exchanges between the different channels and control volumes of the model (Table 2). It is important to highlight that the same model calibrated for this research was used in other studies conducted in the NCM region and near Macapá and Santana (*M. de Abreu et al., 2020*; *Cunha et al., 2021*; *Araújo et al., 2022*).

Hydrodynamically, both in experiments and in numerical simulations in the NCM, a strong influence of the propagation of the tide on the CSA flow was observed when the natural reversal of the total flow was approximately 4 h in a less rainy season (November) and 3.5 h in a rainy season (May). This hydrodynamic behavior has been observed in similar stretches of the Amazon River (*Beardsley et al., 1995*; *Silva, 2002*; *Gallo, 2004*; *Vilela, 2011*; *Nzualo, Gallo & Vinzon, 2018*).

Hydrodynamic characteristics are fundamental for estimating and interpreting, for example, the surface renewal rates of liquid mass (*M. de Abreu et al., 2020*), the dispersive behavior of passive agents in the flow, and self-depurative processes in the Lower Amazon River (*Cunha et al., 2001*; *Cunha et al., 2021*). A relevant factor that directly influences hydrodynamic behavior and pollutant dispersion is the bathymetric characteristics in the region, which are very variable (*Pinheiro et al., 2008*; *Cunha et al., 2012*; *Araújo et al., 2022*). For example, the river is relatively shallow (between 3 and 5 m) in front of Macapá but much deeper in the Santana Channel (up to ≈65 m) (Supplement-S2a–S2b).

Supplement-S4a–S4b shows different current intensity values and their spatial distribution between the different effluent discharge points (P1–P11) and the reference lines (RF-300, RF-600, and CAESA). Nevertheless, the points with the lowest current values were P2 (Channel of Jandiá), CAESA, P6 (downstream of the Igarapé Fortaleza), P8 (downstream of the Matapi River), and P10 (Vila Nova River). Furthermore, the points with the highest current values were P3 (downstream of CAESA) and P5 (downstream of the Igarapé da Fortaleza).

In the flood tide, we detected changes in the current speed intensity at some effluent discharge points. At the CAESA point, we found a significant increase in current velocity ($-0.83$ ms$^{-1}$) compared to the current velocity in the stream ($+0.74$ ms$^{-1}$). This relative difference of 12.16% tends to influence the mass and energy exchange rate of the CSA and NCM in these different tide phases (*M. de Abreu et al., 2020*). In the margins, velocities of the ebb and flooding currents tend to be closer to each other but clearly lower than in the center of the CSA and NCM. Similar behavior has been found in previous studies (*Bougeard et al., 2011*; *Harari et al., 2013*; *Barros et al., 2015*; *Reder, Flörke & Alcamo, 2015*; *Batista & Harari, 2017*).

The results of the Lagrangian model in the $T_1$ decay time suggest that pollutant plumes originating in sources between the Igarapé do Curiaú and the Vila Nova River are more concentrated between P1 and P9 (Figs. 3–4). However, the most significant pollutant concentration peaks were observed in only a restricted section of the domain (P2 and P6). This behavior is very similar to that observed by *Cunha et al. (2012)* at points near the mouth of the Matapi River.

The stretches between P2 and P6 are the most critical concerning bathing and basic sanitation (near P5), where the direct impact of the CSA waters occurs. Between P2 and P4, the CAESA is located in Macapá, downstream of the Macapá Biological Stabilization Pond (Lagoa de Estabilização Biológica de Macapá—LEB), constituting one of the main sources of domestic pollutants in the municipality. Although LEB is the most important infrastructure in Macapá for receiving sanitary sewage of domestic origin, it cannot assimilate all the sewage collected. In addition, it operates without adequate management control, with a compromised and non-quantified level of treatment due to the lack of quantitative and qualitative monitoring of coliforms, BOD, and chemical oxygen demand (COD) concentrations. This makes LEB a significant source of inefficiently treated effluent dumps (Igarapé das Pedrinhas) (*Grott et al., 2018*; *Viegas et al., 2021*).

Similar studies (*Cunha et al., 2004*; *IMAP, 2018*) indicate that Fazendinha Beach (P5) and the Macapá river banks area often have high pollutant concentration peaks, especially in urban drainage channels and stretches close to the riverbanks (*Cunha, 2018*). Therefore, P2, P4, and P5 have the worst water quality indices, representing greater health and ecological-environmental risk and favoring the proliferation of pathogenic bacteria and viruses, such as SARSupplement-CoV-2, by fecal-oral contamination (*Pereira et al., 2014*; *Costa et al., 2020*). In addition, there is an imminent threat to aquatic species and local biodiversity (*Velz, 1984*; *Xu et al., 2013*; *Araújo et al., 2022*).

As observed, the pattern of spatial variation in the concentration of pollutant plumes in dynamic aquatic environments depends on hydrological seasonality (*Barros et al., 2015*). For example, in May and November (Figs. 3A–3B), just one day after the discharge of pollutants into the flood tide, the plume expanded laterally up to 2 km toward the center of the NCM. That is, the simulated plumes compromised and exceeded the reference zone of influence (CAESA, RF-300, and RF-600). The only exception occurred three days after launch (Figs. 4C–4D).

It is worth emphasizing that the flow of the Amazon River significantly reduces in November (dry period) (*Cunha et al., 2017*; *M. de Abreu et al., 2020*). In this period, there is also a significant and proportional increase in wind intensity in the NE direction (Figs. 2A–2B), favoring the retention and lateral increase of the pollutant plume. Thus, frequent pollutant concentration peaks occur in some dispersion time intervals. Previous studies have detected similar effects (*Dunn, Zigic & Shiell, 2014*; *Tang et al., 2014*; *Barros et al., 2015*; *Batista & Harari, 2017*).

The increasing contribution of local pollutants close to CSA and NCM also represents an increase in nutrient concentration, favoring the proliferation of green or blue algae, such as cyanobacteria, which produce dangerous toxins (eutrophication) (*Viegas et al., 2021*). A recent study detected microcystin-LR for the first time in the Amazon River and
in the water treatment plant of Macapá (*Oliveira et al., 2019*). Therefore, the quality of raw and treated water from the CAESA may already be progressively compromised.

Pollution by domestic sewage can increase the incidence of parasites in fish, potentially impacting aquatic life around pollutant sources (P2–P6), interfering in hyporheic zones (interface between the water column and the upper layer of sediments) and the balance of biogeochemical reactions (*Abreu & Cunha, 2015*; *Tavares-Dias et al., 2015*; *Jenerowicz & Walczykowski, 2015*; *Abreu & Cunha, 2016*). Therefore, the reference lines RF-300 and RF-600 are helpful in the analysis of dispersive processes of domestic effluents, compared to similar situations in other studies (*Velz, 1984*; *Ben Hamza et al., 2015*; *Van Gorder et al., 2015*).

Hydrological seasonality significantly influenced the spatiotemporal variations of pollutant concentrations at the CAESA reference target sites (RF-300 and RF-600). As expected, in the $T_1$ and $T_3$ time scenarios, the overall average concentration of pollutants in May (higher dilution capacity and renewal rate) had lower values than in November (inverse behavior) (*M. de Abreu et al., 2020*). This can be explained by the simultaneity of hydrodynamic (currents, water volume, reduced transport capacity, dilution) and meteorological effects (increased wind intensity–longitudinal compression and lateral expansion of the plumes in the NCM and CSA) (Figs. 3 and 4).

In $T_2$, the average concentrations were higher in May in RF-300 and RF-600, indicating the relative importance of the decay kinetics of the pollutant and impacting the rates of organic matter decomposition, which may affect the ecological balance, food chain of the aquatic biota and sanitary legal standards within the spatial domain of the area (*Velz, 1984*; *Roversi, Rosman & Harari, 2016*; *Von Sperling, 2017*; *Cunha et al., 2021*). Also, at $T_2$, we detected a trend of peak concentrations at the CAESA reference site. This suggests that this region is vulnerable and sensitive, posing a high risk to the environment and public health, both upstream and downstream. For example, if pollution rates maintain this trend, the risks and operational costs of the water collection and treatment system may be compromised and elevated (*Velz, 1984*; *Von Sperling, 2017*; *Oliveira et al., 2019*; *Oliveira et al., 2014*).

Therefore, the spatial variation of the concentration of the plumes is intense close to the margins (or the point and diffuse sources of pollution), diminishing toward the center of the channels or moving longitudinally along the margin but often concentrated and limited to the imaginary line RF-600. Paradoxically, in these scenarios, plumes tended to have lower average concentrations in these locations. This was found in all simulations and at points closer to the riverbank (RF-300). This fact corroborates the Technical Standard [ABNT NBR 6022 (2003)], which recommends distances of at least 300 m beyond the margin when designing underwater outfall facilities (*Rodrigues, 2012*).

Part of this variation can be explained by the geomorphological characteristics of the coast between the Vila Nova River and the Igarapé do Curiaú. In this stretch, there are significant changes in the patterns of currents, which favor the longitudinal displacement of the plumes throughout the tidal cycles, especially in relation to the margins (RF-600). Depending on the tide phase or the seasonal period, plumes tend to concentrate naturally closer to the margins (RF-300). This makes the distance from the margin a critical design

parameter for the final discharge of sanitary sewage, such as underflow outfalls (*Cunha, 2018*).

In November, plume behavior may be more closely correlated with the intensity and preferential direction of the wind in Macapá and Santana (NE direction) since the higher intensity during this period favors retention and lateral expansion of the plume, significantly influencing the lateral dispersive process, when evaporation rates may increase the effects of pollutant concentrations (*Cunha, 2018*). As a consequence, a longer residence time for pollutants (Tr) near the banks of the channels (*M. de Abreu et al., 2020*) reduces dilution capacity (*Cunha et al., 2001*) and increases the interactions associated with demographic dynamics and the volume of urban sewage produced in time and space (*Velz, 1984*; *Von Sperling, 2017*).

Hydrodynamically, there is a hypothetical "hydraulic barrier" that can favor/hinder the dispersive process (lines of preferential and intense currents in the longitudinal flow direction to the detriment of lateral recirculating currents) (*Cunha et al., 2021*). Such a "hydraulic barrier" seems to hinder the lateral dilution process of pollutants while tending to keep pollutant concentrations within the limits of RF-300 or RF-600 or as otherwise imposed by CONAMA standard 274/2000 (P5–P6). This fact may help explain the reasons why these waters tend to have limited swimming at these distances since they pose health risks to swimmers due to contact with vectors that cause waterborne diseases (*Campos & Cunha, 2015*; *Von Sperling, Verbyla & Oliveira, 2020*; *Viegas et al., 2021*). In addition, in the region, there are reports of non-compliance with swimming recommendations, particularly during the high season in tourism (*Damasceno et al., 2015*), in zones between the coast and reference lines RF-300 and RF-600.

The simulated scenarios suggest that the final discharge of pollutants in Macapá and Santana significantly compromises water quality despite the immense self-depuration capacity of the Amazon River (*Cunha et al., 2001*). In these terms, we reject the hypothesis of "unlimited" capacity for dilution and self-depuration in this stretch of the water body, given the high loads of pollutants dumped without treatment, which has compromised water quality parameters for decades (*Cunha et al., 2004*; *Cunha et al., 2005*; *Damasceno et al., 2015*). That is, there would be an "unlimited" capacity to biodegrade these quantities of pollutant loads from Macapá and Santana if the plumes disperse further to the center of the channels (*Cunha et al., 2001*; *Cunha et al., 2005*).

The simulations indicate that, regardless of the volume of the Amazon River's water, current loads will always tend to concentrate close to the margins of the CSA and NCM, up to an imaginary limit of 600 m from the coastline, compromising the entire length of the Macapá and Santana riverbanks. The simulations also suggest that during flood tide phases, the plumes can expand toward the municipality of Mazagão Velho (upstream) in favorable hydrological periods (*e.g.*, Fig. 3B).

In this respect, the simulations were important to assess the health impacts caused by point sources and diffusing pollution from inadequately treated domestic sewage (*Oliveira et al., 2014*). Therefore, the present study is very promising for analyzing the design of future sanitary infrastructure and water resource projects, as well as benefiting conservation of

aquatic biodiversity in the tropics to improve sewage system design projects with significant positive impacts on public health in Macapá and Santana.

We believe that our findings on the hydrodynamic behavior of the Amazon River and its influence on the behavior of sewage plumes will be useful in procedures that support eminently sustainable decision-making for current and future infrastructure projects in the coastal region. We also hope these procedures will benefit ecosystem conservation management that lacks scientific knowledge on sanitation, which has particular environmental, social, and economic importance in the Amazon.

## CONCLUSIONS

We simulated scenarios of dispersion of domestic pollutant plumes along the coast of Macapá and Santana in Amapá, resulting in a reliable representation of their spatial and temporal variations, extensions, and peak concentrations in strategic locations of interest to basic sanitation. This unprecedented analysis represents an approximation of the probable impacts of domestic sewage pollutants along the coast, supporting our hypothesis. The simulation model proved robust and capable of representing, in a first approximation, the extent and level of the effects and impacts of the main point and diffuse loads on the water quality of the most urbanized coastal-estuarine region of Amapá.

The simulated scenarios suggest a strong dependence on hydrodynamic characteristics (seasons and tides), location of pollutant sources, kinetic pollutant properties, and geomorphological and meteorological characteristics, such as wind intensity and seasonal direction. The place indicated by the ETA of the CAESA was considered the most critical hotspot due to the frequent increase in pollutant concentration, with eventual plume recirculation in the vicinity. Therefore, there are considerable risks that these sources affect the water collection system, as it is a sensitive and relevant receiver for sanitary actions to combat and reduce pollutants.

The main numerical outputs showed concentration peaks that could be aggravated or reduced depending on the decay kinetic characteristics of the pollutants ($T_1$, $T_2$, and $T_3$), which could significantly affect the position and period of maximum concentrations of the plumes in the CSA and NCM sections.

The hypothesis of unlimited self-depuration of the Amazon River was rejected. Despite their remarkable dilution capacity, plumes tend to concentrate closer to the margins or longitudinally close to the edge. This fact suggests that the "available volume and liquid discharge" of the Amazon River in these regions is insufficient to self-clean the simulated loads because the plumes tend to concentrate close to the banks, where the waters also tend to be shallower up to a distance of RF-300 m or RF-600. Thus, the model suggests that the surface renewal rates in these regions tend to be lower than the center of the channels. However, dispersion processes may intensify or weaken during hydrological periods when the wind influence is more significant.

In this regard, our study significantly contributes to the basic sanitation of Amapá, a Brazilian state lacking sewage systems and efficient sewage treatment. Therefore, the present study can be used as a tool for planning, managing, and directing preventive actions to

contain the current pollution level due to domestic sewage in the urban areas of Macapá and Santana. The main applications of these results would be: (a) design and implementation of sanitary sewage treatment plants (Macapá and Santana); (b) alternative studies for constructing subfluvial emissaries in natural channels (mainly in NCM) or more distant and safe places for the edges, avoiding the diffuse dispersion that is difficult to control and monitor, whose solution would benefit the multiple uses of water in these fragile coastal environments; (c) assist planning/monitoring water quality under the influence of the impacts of pollutant plumes on the aquatic ecosystems of this stretch of the Amazon River, aiming at gradual universalization of sanitary sewage indicators.

However, the present investigation has limitations. The evaluation of numerical dispersion scenarios in intermediate hydrological periods, which was not performed here, is necessary, allowing a new model calibration process. This could be overcome by adding more experimental campaigns of bathymetric, hydrodynamic, and water quality surveys to extend the series of data available for this type of study. In fact, the process of monitoring water quality and calibrating the model would be more efficient, aiming at the model's future validation in generating new plume dispersion scenarios. The impacts would be positive for the water resources sector, conservation of coastal ecosystems, sanitation, and waterway transport.

### Funding
This project was supported by CNPq: Project No. 309684/2018-8 and 314830/2021-9, TEDPLAN Project (2018)/FUNASA—UNIFAP. The funders had no role in study design, data collection and analysis, decision to publish, or preparation of the manuscript.

### Grant Disclosures
The following grant information was disclosed by the authors:
CNPq: 309684/2018-8, 314830/2021-9.
TEDPLAN Project (2018) / FUNASA - UNIFAP.

### Competing Interests
The authors declare there are no competing interests.

### Author Contributions
- Carlos Henrique Medeiros de Abreu conceived and designed the experiments, performed the experiments, analyzed the data, prepared figures and/or tables, authored or reviewed drafts of the article, and approved the final draft.
- Elizandra Perez Araújo analyzed the data, prepared figures and/or tables, and approved the final draft.
- Helenilza Ferreira Albuquerque Cunha conceived and designed the experiments, performed the experiments, analyzed the data, prepared figures and/or tables, authored or reviewed drafts of the article, and approved the final draft.

- Marcelo Teixeira conceived and designed the experiments, performed the experiments, authored or reviewed drafts of the article, and approved the final draft.
- Alan Cavalcanti da Cunha conceived and designed the experiments, performed the experiments, analyzed the data, prepared figures and/or tables, authored or reviewed drafts of the article, and approved the final draft.

## Data Availability

The actual elevation data (available in the Real Tide part of the Supplemental File) is available from the IBGE website (https://www.ibge.gov.br/geociencias/informacoes-sobre-posicionamento-geodesico/rede-geodesica/10842-rmpg-rede-maregrafica-permanente-para-geodesia.html).

The other data, such as simulated tidal elevation and river net discharge, which were measured in the field and produced by the model, are results generated by the study and available in the Supplemental File. These data were obtained through the simulation of SISBAHIA and field measurements using ADCP.

## Supplemental Information

Supplemental information for this article can be found online at http://dx.doi.org/10.7717/peerj.16933#supplemental-information.

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
