# Peer review of "Domestic sewage dispersion scenarios as a subsidy to the design of urban sewage systems in the Lower Amazon River, Amapá, Brazil"

_PeerJ, doi:10.7717/peerj.16933_

## Round 0.1 · original submission · Major Revisions

We have recieved three reviews for your paper, two rather short and one more in-depth. The reviews agree that the topic is relevant and important, particularly for the region due to a lack of prior studies. However, the manuscript needs several improvements before it can be considered for publication. Please see the reviewer comments, but particularly note the need for (1) improved English and better focus in the paper, (2) clearer figures, (3) additional information about the modelling methods, (4) experimental design and need for additional scenarios, and (5) objectivity in the presentation of the results.

**Language Note:** The Academic Editor has identified that the English language must be improved. PeerJ can provide language editing services - please contact us at copyediting@peerj.com for pricing (be sure to provide your manuscript number and title). Alternatively, you should make your own arrangements to improve the language quality and provide details in your response letter. – PeerJ Staff

·

Basic reporting

No comment

Experimental design

- More information about bathymetry in shore;
- more information about the bathymetry of the Amazon River;
- More information about hydrodynamic model in SisBaHiA.

Validity of the findings

The study used a well-established tool in hydrodynamic studies in estuarine regions and reservoirs. The scenarios evaluated are hypothetical.

Additional comments

Plumes tend to concentrate closer to the margins when pollutant release on the shore. Therefore, new scenarios are suggested: Between P1 through P5, with sewage discharge away from the shores, evaluate the dispersion of pollutants. Objective: to verify if the pollutant plume enters the central region of the Amazon River, minimizing the impact on the Vila Nova River and in the CAESA ETA.

Reviewer 2 ·

Basic reporting

review the quality of the generated images.

Experimental design

"no comments"

Validity of the findings

The discoveries are essential for the Amazon region, since the region is still very far from the desired references for sanitary sewage and the study emphasizes this need and the consequences of its lack.

Additional comments

Manuscript needs some writing revisions, and sharpening of images.

Reviewer 3 ·

Basic reporting

• The English needs a thorough revision. Too much direct translation of Brazilian Portuguese, which results in unusual English sentences. There are units mixing Portuguese in one sentence and English in following sentences, for instance, in line 214 one sees NMP/100mL, and it should be CFU/100mL.
• The intro and background context should be more focused on the region of interest. There is generic information regarding Brazil as a whole, which is not necessary.
• The structure follows the Peerj standards but deserves more conciseness and objectivity. Literature is adequate.
• The figures must be improved. For instance, maps of isolines of concentration showing similar results, for different scenarios or times, use different color scales. They must use the same color scale to facilitate comparisons - for the case, use of a log color scale is appropriate. The map showing the computational finite element mesh is quite inadequate, as it shows a mesh that is much coarser than the real one. The problem is that the model uses quadratic finite elements, and the authors have presented a map showing only the vertices of the elements. For instance, quadrangular elements have 9 nodes, the presented map only shows the four vertices of each element.
• Raw data should clearly show which values were used to calibrate the hydrodynamic model. All the authors say is that they have used a pre calibrated model. At least one graph comparing modeled vs. measured water levels should be presented. In addition, they should also show a comparison of measured vs. modeled discharges. Both, with proper analyses.

Experimental design

The whole description is quite insufficient, and there are unacceptable inadequacies. For instance:
1. The modeling domain is not defined. Where are the boundaries?
2. The setting up of the hydrodynamic scenarios are not defined. In lines 163-165 it is written “…analysis of predicted tide and hydrodynamic simulation (for May, August, and November), as described in Abreu et al., 2020.” Three months are mentioned, the days of each month are not, nor the corresponding year!
3. The hydrodynamic model conditioning is not explained. What were the boundary conditions for each scenario? Has meteorological data, such as wind, been used? Etc.
4. The setting up of the Lagrangian Transport Model is not explained. The authors used a constant T90 value which is quite precarious in analyses of bacterial decay. In the Lagrangian Transport Model of SisBaHiA there is a very sophisticated bacterial decay model, with T90 varying according to water temperature and salinity, and mainly as a function of solar radiation. To use a constant value of T90 is unacceptable in this kind of analysis.
5. Maps presenting isolines of CFU/100mL at selected times is far from being the best way to analyze this kind of modeling results. It is way more effective to present maps showing isolines of the percentage of time where the plumes from different polluting sources have passed with concentrations above a given limit. And many more options.

Validity of the findings

The validity is quite questionable, for the reasons exemplified above.

Additional comments

The subject is rather relevant and deserves to be better developed and improved for publication.

---

## Round 0.2 · accepted · Accept

Thank you for submitting a revised manuscript, which was greatly improved following the reviewer inputs. I acknowledge that you were not able to produce the additional isolines map but feel that the results are sufficiently well presented. Actually, I consider that your work is excellent and I look forward to seeing its continued development, perhaps expansion to other areas such as Manaus, or applications such as sewage interaction on the floodplain, issues with fisheries, etc. Overall, I wish you congratulations on your research.

One suggestion: some of the supplementary figures would sit well within the main body of the paper, and I suggest you consider this as part of the final copy editing. Thank you for including your data as part of your publication.

·

Basic reporting

Requests in round 1 have been made

Experimental design

Requests in round 1 have been made

Validity of the findings

The study used a well-established tool in hydrodynamic studies in estuarine regions and reservoirs. The scenarios evaluated are hypothetical.

Additional comments

Requests in round 1 have been made

Reviewer 2 ·

Basic reporting

Bibliographic references, sufficient field history/context provided.
Structure of professional articles, figures, tables. Shared raw data.

Experimental design

The Research Question was very well defined, relevant and meaningful. showing that the research fills an important knowledge gap for the study region.

Validity of the findings

Conclusions are well formulated, linked to the original research question, and limited to supporting the results.

Additional comments

Important research that paves the way for this line of study in a potentially sensitive region that is the Amazon, showing the importance and relevance of the research.